# Comprehensive Evaluation of Unsupervised Image Enhancement for Volumetric Fetal Brain MRI

**Yingqi Hao**[*1]                                     HAO-YQ22@MAILS.TSINGHUA.EDU.CN
**Mingxuan Liu**[*1]                                          ARKTISX@FOXMAIL.COM
**Hongjia Yang**[1]                                   YANGHJ23@MAILS.TSINGHUA.EDU.CN
**Haoxiang Li**[1]                                   LIHAOXIA24@MAILS.TSINGHUA.EDU.CN
**Xuguang Bai**[1]                                      BXG21@MAILS.TSINGHUA.EDU.CN
**Yi Liao**[2]                                                CONNIE0064@126.COM
**Haibo Qu**[2]                                             WINDOWSQHB@126.COM
**Qiyuan Tian**[†1]                                      QIYUANTIAN@TSINGHUA.EDU.CN

[1] *School of Biomedical Engineering, Tsinghua University*

[2] *Department of Radiology, West China Second University Hospital, Sichuan University*

**Editors:** Accepted for publication at MIDL 2025

## Abstract

MRI provides superior soft tissue contrast over ultrasound, making it essential for evaluating fetal brain development and pathology. Although the recently proposed foundation model BME-X represents the first dedicated approach for fetal MRI enhancement, its generalizability to heterogeneous clinical datasets remains unproven. To bridge this gap, we conduct the first comprehensive comparison of BME-X and other unsupervised image enhancement methods on normal and pathological fetal brain MRI, based on tissue segmentation accuracy, tissue contrast t-score (TCT), lesion fidelity, and reader assessment. Results show that a pre-trained 3D convolutional variational autoencoder (VAE) achieves more effective enhancement compared to BME-X. Code and pre-trained weights are available at https://github.com/yingqihao2022/FetalBrainEnhancement.

**Keywords:** Fetal brain MRI, Image enhancement, BME-X, Foundation model, 3D VAE.

## 1. Introduction

Structural MRI's non-invasive safety, superior soft tissue contrast, and radiation-free nature make it ideal for fetal brain characterization (Sun et al., 2024; Hao et al., 2025; Sanchez et al., 2023; Gholipour et al., 2011). However, 2D thick-slice T2-weighted acquisitions are vulnerable to motion interference from fetal movements and maternal respiration, causing blurring/ghosting that complicates neuroimaging analysis (Liu et al., 2025; Xu et al., 2020). While reconstruction tools like NiftyMIC (Ebner et al., 2019) and NeSVoR (Xu et al., 2023) can convert motion-corrupted 2D stacks into 3D volumes, persistent artifacts and noise degrade output quality (Sun et al., 2024).

Enhancing volumetric fetal brain MRI faces two key challenges: the lack of noise-free ground truth, which limits supervised approaches, and the presence of non-traditional noise types, making unsupervised methods like Noise2Void (Krull et al., 2018), Zero-shot

---

[*] Contributed equally

[†] Corresponding author

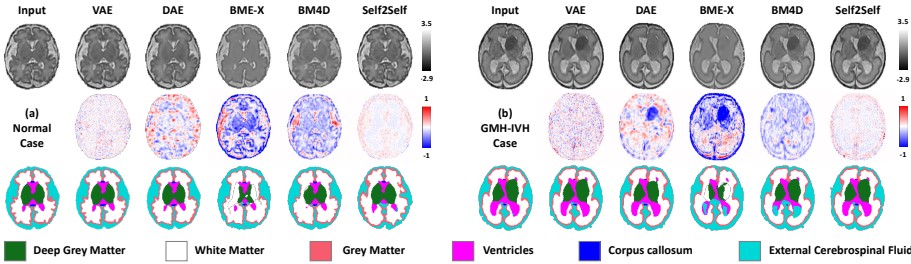

Figure 1: Examples on a normal fetus (a) and a GMH-IVH fetus (b). The first row: axial slices of input/output images. The second row: residual maps (input - output). The third row: tissue segmentation results.

Noise2Noise (Mansour and Heckel, 2023), and Self2Self (Quan et al., 2020) less effective. Recently, Sun et al. introduced BME-X (Sun et al., 2024), the first successful model designed for this task, aiming to improve image quality by sharpening details and unifying tissue intensity ranges. While advertised as applicable to brain MRIs across all ages, their experiments focus on neonatal and adult cases, with limited validation on fetal images. To address the above problems, this study evaluates unsupervised image enhancement models for volumetric fetal brain MRI for the first time, including both normal and pathological cases. Evaluation includes tissue segmentation accuracy, the tissue contrast t-score (TCT) (Pizarro et al., 2016), lesion fidelity, and reader assessment. Results show that the pre-trained 3D convolutional VAE achieves state-of-the-art (SOTA) image enhancement, excelling in preserving brain structures and lesion fidelity.

## 2. Methods

### 2.1. Data Acquisition and Preprocessing

The study used MRI data from 442 pregnant women, with 432 normal fetal brains and 10 brains with Germinal Matrix Hemorrhage and Intraventricular Hemorrhage (GMH-IVH) lesions. 2D T2-weighted TSE imaging data were acquired in axial, coronal, and sagittal directions. The NeSVoR (Xu et al., 2023) method was used for slice-to-volume motion correction and 3D volumetric reconstruction.

### 2.2. Competing Methods

Five unsupervised image enhancement methods were implemented for fetal brain MRI: (1) 3D Convolutional VAE: Previous studies (Prakash et al., 2021) have explored direct denoising with VAE in microscopic imaging. The VAE model is designed to sample diverse solutions from trained posteriors by directly predicting per-pixel mean and variance, enabling the potential to learn right values. In this study, the MONAI framework (Pinaya et al., 2023) is utilized to implement a 3D VAE. (2) DAE: The denoising autoencoders (DAEs) eliminate noise through trained reconstruction of clean data from corrupted inputs. Our implementation adopts the DAE framework established in prior research (Hao et al., 2025). (3) BME-X: A recently proposed foundation model for brain MRI image enhancement (Sun

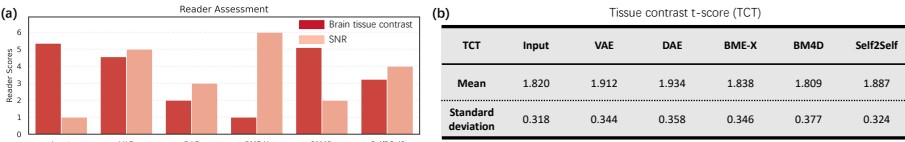

Figure 2: Reader assessment and TCT test results from different methods.

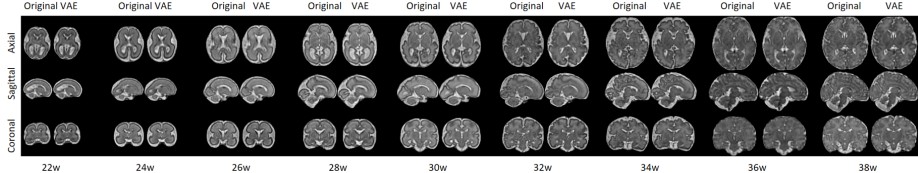

Figure 3: Enhanced results of the VAE model for brain volumes at various GAs (22w-38w).

et al., 2024). (4) BM4D: BM4D is an algorithm for attenuation of additive spatially correlated stationary (aka colored) Gaussian noise for volumetric data (Maggioni et al., 2013). (5) Self2Self: A self-supervised denoising technique using internal image statistics without external training data (Quan et al., 2020).

The VAE and DAE models were trained on 390 normal volumetric fetal brain MRI images. BME-X used its original pre-trained weights. Self2Self and BM4D are unsupervised single-image methods requiring no additional training data. The remaining 42 normal and 10 GMH-IVH fetal brain MRI volumes were reserved for testing.

## 3. Results and Conclusion

Figure 1 shows the enhancement results of normal and GMH-IVH fetal brain MRI images using different methods. Key observations include: (1) Residual maps reveal that only VAE effectively removed noise while preserving normal tissue contrast, while other methods changed tissue contrast during denoising, as confirmed by segmentation results (using a pretrained model (Fidon et al., 2024)); (2) Only VAE, BM4D, and Self2Self preserved hemorrhagic lesions in GMH-IVH brains, ensuring lesion fidelity. Figure 2 (a) presents reader assessment by an experienced radiologist (Y.L.), showing VAE's balanced performance in signal-to-noise ratio (SNR) and tissue contrast. Figure 2 (b) illustrates TCT metrics (calculated following (Sun et al., 2024)), with VAE and DAE achieving top scores. Figure 3 illustrates VAE's enhanced results across various gestational ages (GAs), demonstrating its generality. In conclusion, VAE is the SOTA volumetric fetal brain MRI enhancement model. Future work should explore VAE-based models to improve fetal brain image enhancement.

## Acknowledgments

Funding was provided by the Tsinghua University Startup Fund, Dushi Program (grant number 20241080026) and Initiative Scientific Research Program.

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
