# OpenReview forum: "Comprehensive Evaluation of Unsupervised Image Enhancement for Volumetric Fetal Brain MRI"
_MIDL.io/2025/Short_Papers — MIDL 2025 - Short Papers_

### Official Review · Reviewer_BfcL · 2025-04-22

**Rating:** 4
**Confidence:** 4

**Summary:**

The authors propose to compare different methods to post-process 3D fetal brain MRIs that have been reconstructed from thick 2D slices. Specifically, the authors evaluate the ability to remove noise/artifacts while keeping a good tissue contrast. This task is particularly difficult for fetal MRI, becausefetal and respiratory motions  prevent the acquisition the acquisition of noise-free ground-truth volumes.
Here the authors find that a VAE obtains the best results, by using reader assessment and a tissue contrast score.

**Strengths:**

- Clear objectives and contributions (benchmarking of existing methods)
- Interesting area of research, which is under-represented
- Well-written paper
- Interesting outcomes

**Weaknesses:**

- Lack of explanations about the metrics (which is the central part of this work): a) the TCT metric is not well-known and is not explained at all. B) the “reader assessment” metric is not very rigorous as it seems it was performed globally (as opposed to per-case assessments), and the meaning of the ratings is not explained.
- Results are stated but not commented. For example the zero-shot methods (especially Self2Self) seems to be working very well despite it does not need any training.
- Missing information about the images (sequence, resolution, size, scanner).
- Why not reporting Dice scores?
- The testing set is too small compared to the training one (12% only), and there is no mention of a validation set.

---

### Decision · Program_Chairs · 2025-05-01

Accept